# A Rare Case of a Primary Leiomyoma of the Clivus in an Immunocompetent Patient and a Review of the Literature Regarding Clival Lesions

**DOI:** 10.3390/diagnostics13010009

**Published:** 2022-12-21

**Authors:** Jacek Kunicki, Natalia Rzewuska, Michał Kunicki, Piotr Wiśniewski

**Affiliations:** 1Department of Neurosurgery, Maria Sklodowska-Curie National Research Institute of Oncology, 02-781 Warsaw, Poland; 2Department of Gynecological Endocrinology, Medical University of Warsaw, 00-315 Warsaw, Poland; 3INVICTA Fertility and Reproductive Center, 00-019 Warsaw, Poland; 4Department of Pathology and Laboratory Diagnostics, Maria Sklodowska-Curie National Research Institute of Oncology, 02-781 Warsaw, Poland

**Keywords:** primary leiomyoma, clivus, clival diseases, chordoma, fibrous dysplasia

## Abstract

Leiomyomas are common lesions that are usually located in the genitourinary and gastrointestinal tracts. Primary leiomyomas at the skull base are uncommon. They are composed of well-differentiated smooth muscle cells without cellular atypia. The diagnosis of a leiomyoma has to be confirmed by immunohistochemistry. The tumor tissue is immunoreactive for SMA, S100 and cytokeratin. Leiomyomas mainly occur in immunocompromised patients. Most tumor tissues are positive for EBV. The presented case is that of a 56-year-old immunocompetent woman with a tumor on the clivus. The radiological images suggested chordoma or fibrous dysplasia. Transnasal transsphenoidal surgery was performed. The tumor tissue consisted of well-differentiated smooth muscle cells with elongated nuclei. Immunohistochemistry revealed a positive reaction for desmin, SMA and h-Caldesmon and a negative reaction for S100, beta-catenin, PGR and Ki67. The leiomyoma diagnosis was subsequently established. To the best of our knowledge, the case of a primary leiomyoma on the clivus of an immunocompetent patient is the first to be described. We also extensively reviewed the literature on the immunohistopathological and radiological differential diagnosis of clival lesions.

## 1. Introduction

Leiomyomas are benign tumors that are typically located in the genitourinary and gastrointestinal tracts, although the much less common hepatic leiomyomas are also possible [1,2,3]. Intracranial leiomyomas are a rare find in contrast to the frequent occurrence of leiomyomas in the uterus [2].

Uterine leiomyomas, known as fibroids, are among the most common benign tumors in women and are diagnosed in up to 70% of Caucasian women and up to 80% of African women who are 50 years of age [4]. Uterine fibroids are the most common indication for hysterectomy in the United States [5]. The etiology and initiating factors of these lesions are unknown. Estrogen and progesterone stimulate their growth and recently, the role of environmental estrogens has also been identified. Additionally, several risk factors have been reported, such as early menarche, late reproductive age, African American ethnicity, obesity, smoking, dyslipidemia, hormone replacement therapy (HRT) and positive family history [5,6]. Finally, genetic factors can also affect the etiopathology of leiomyomas [7]. Karyotypic abnormalities have been found in 40% of examined fibroids. There are some growth factors and receptors that are elevated in the presence of fibroids, such as transforming growth factor-β3 (TGF)-β, epidermal growth factor (EGF), basic fibroblast growth factor (bFGF) and insulin-like growth factor-I (IGF) [5].

Leiomyomas consist of well-differentiated myocytes with small components of proliferating cells that have minimal metastatic potential [8]. These tumors progress to malignant features in fewer than 0.1% of cases [9]. The tumorigenesis of intracranial leiomyomas remains unexplained, but it is assumed that they originate from leptomeningeal coatings that contain leptomeningeal pluripotent mesenchymal stem cells from the dura mater or from epithelium in cerebral blood vessels. Most of the scientific reports regarding leiomyomas on the skull base and intracranial leiomyomas are case reports [10] and the first ever case was presented by Kroe et al. in 1968 [11]. Only about 30 cases of primary intracranial leiomyomas have been described [12]. Intracranial leiomyomas are mainly located in the cerebral, medial fossa and sellar regions, although intraventricular lesions have also been reported [13]. The radiological and clinical features of these tumors are nonspecific and require confirmation via histopathology [2]. The majority of intracranial leiomyomas occur in immunocompromised patients, including those with human immunodeficiency virus (HIV) and transplant recipients, but leiomyomas can also occur in immunocompetent patients. The simultaneous occurrence of Epstein–Barr virus (EBV) infection has also been observed. EBV has been associated with smooth muscle tumors in immunocompromised patients because it is capable of infecting them [1,10,12,13,14]. Leiomyoma treatments include surgical resections and subsequent radiological follow-ups [8].

## 2. Case Report

A 56-year-old non-obese, non-smoking female (gravida one, para one) presented with severe diffuse headaches lasting for 2 months, accompanied by retro-orbital pain. She had no history of uterine fibroids according to sonography (Figure 1) nor had she had hormonal treatment after menopause or a hysterectomy. There were no abnormalities found during physical and neurological examinations. There was no pathological resistance nor tumors in the subcutaneous tissue upon palpation. The basic laboratory tests were within the normal ranges. The patient’s anti-HIV tests were negative. She had no history of neoplasm nor recurrent infections and she did not use steroids. There were no medical conditions that indicated immunodeficiency.

The patient reported nonspecific signs of fatigue and impaired concentration. The magnetic resonance imaging (MRI) of the head revealed a 16 × 18 × 12 mm tumor in the region of the clivus. On T1-weighted imaging, the lesion was isointense and peripherally enhanced after contrast administration. The borders of the lesion were hyperintense while the main part was isointense (Figure 2A). On T2-weighted imaging, the lesion was isointense as well (Figure 2B). On computed tomography (CT), the tumor had smooth outlines with a hypersclerotic rim. It was located centrally and did not show any penetration into the sinuses (Figure 2C). In diffusion-weighted imaging (DWI), no signals of recent ischemic changes were found. The radiology images were also nonspecific and suggested chordoma or fibrous dysplasia.

Endoscopic transnasal transsphenoidal surgery was performed, with the complete resection of the lesion. An intraoperative macroscopic image of the tumor showed a well-circumscribed, solid mass that was focally eroding the sclerotic bone margin within the clivus.

Histopathologically, there were non-atypical spindle cells with no mitotic activity or necrosis (Figure 3A).

The immunohistochemistry reactions for the S100 protein, beta-catenin (Figure 3C), cytokeratin AE1/AE3, CD34 and progesterone receptor (PGR) were negative. The cell nuclei were negative for Ki-67, confirming their low proliferative potential. These results were typical of a benign tumor with smooth muscle differentiation; thus, the diagnosis of an intraosseous leiomyoma was established. Immunohistochemistry confirmed the absence of EBV in the pathological tissue.

Postoperative MRIs showed minimal residual lesions within the sphenoid sinuses (Figure 4A–C). The 4-year follow-up of the patient revealed no clinical or radiological features of tumor recurrence. 

## 3. Discussion

The presented case was an uncommon localization of a leiomyoma. The typical locations for these lesions are the uterine or gastrointestinal tracts and they rarely metastasize in the CNS [10]. In the literature, similar primary cases have been described in the sellar and parasellar, temporal, middle fossa, intraventricular and cavernous sinus regions, in order of the frequency of occurrence [9,12]. Single cases concerning the basal ganglia and occipital lobe have also been documented [9,12,13]. The clivus is one of the rarest locations for these tumors [9,15]. 

The main manifestations of this pathology include headaches and increased intracranial pressure. Less common symptoms include cranial nerve paralysis and focal neurological deficits [13]. The age distribution is as follows: the youngest reported patient was 3 years old [16] while the oldest was 68 years old [12]. So far, primary leiomyomas have occurred in 18 women and 14 men [13,16].

To the best of our knowledge, 32 cases among the adult population have been described in the literature to date. Of those, 16 patients had immunodeficiencies, such as HIV or being transplant recipients, but the rest of them had intact immune systems or unknown immune status [12,13,16]. This pathology has also been reported in young and immunocompetent patients (aged between 17–50) and for this reason, immune status is not strongly believed to be fully correlated with the occurrence of primary intracranial leiomyomas [11]. 

This pathology in the clival region has only been reported in an HIV-positive 31-year-old women [15]. Our patient did not have any abnormalities in her medical history nor were deviations found in physical examinations or laboratory tests that might indicate impaired immunity. She was not HIV positive either. For this reason, we believe that this 56-year-old patient had the first case of a clival leiomyoma in an immunocompetent patient. 

There is also an association between the Epstein–Barr virus (EBV) and the presence of primary leiomyomas; however, the entire pathogenesis of smooth muscle tumors in this location is not understood. The special target for EBV is the lymphocyte B receptor CD21, which is also sometimes present in smooth muscle tumor cells. However, it is unclear whether EBV plays a role in the development of smooth muscle tumors in the cranial location [16]. As the case of our patient shows, intracranial leiomyomas may be EBV-negative and may occur in immunocompetent patients.

Radiologically, the CT and MRI images of leiomyomas are nonspecific [8,12,17]; therefore, accurate differential diagnoses are needed. The treatment of choice is gross total resection and the outcome of this procedure is satisfactory [8]. 

In the next section, we present the radiological and histopathological differential diagnosis of our patient that excluded more common clival lesions.

### 3.1. The Histopathological Differential Diagnosis

Due to the morphological similarities to other spindle cell neoplasms and non-neoplastic lesions deriving from spindle cells, the necessary differential immunohistochemical staining reactions were performed to establish the diagnosis of the leiomyoma. Smooth muscle differentiation in leiomyomas is confirmed by positive staining for smooth muscle actin (SMA), h-Caldesmon, HHF35 and desmin, while immunohistochemical staining for CD34 and S100 protein is negative [18].

The differential diagnosis based on immunohistochemical analyses included benign peripheral nerve tumors, commonly known as schwannomas, and neurofibromas, which are usually positive for S100 protein. Non-ossifying fibromas (NOFs) were also considered. NOFs are composed of spindle cells, frequently accompanied by histiocytes and sometimes by giant multinucleated cells (i.e., cells that do not appear in the images of leiomyomas). In addition, there is no expression of SMA, HHF35 or desmin in non-ossifying fibroma tissue, which aided in making the diagnosis.

The differential diagnosis also suggested a bone desmoplastic fibroma, which produces similar microscopic images and immunoprofile to those of a leiomyoma. However, in those cases, immunohistochemical staining for beta-catenin can clarify the diagnosis as it is positive for desmoplastic fibromas but negative for leiomyomas [19]. 

The differential diagnosis of a spindle cell hemangioma was included. These tumors are characterized by the proliferation of spindle cells and are sometimes accompanied by cavernous blood vessels. Spindle cell hemangiomas are usually found in the skin, but intraosseous occurrence has also been described [20]. The presence of CD34 immunoreactivity in the spindle cells determines the diagnosis of spindle cell hemangioma [21].

Another neoplasm with a similar morphology to that of a leiomyoma is interosseous meningioma. These tumors are positive for cytokeratin and epithelial membrane antigen (EMA), as well as the usual (in about 60–70% of cases in women) expression of the progesterone receptor [22]. Leiomyomas are negative for cytokeratin and EMA, which helped in the differential diagnosis.

Differentiation between metastases and primary leiomyosarcomas was clinically relevant. The presence of cellular atypia and/or high mitotic activity and/or tumor necrosis indicates malignant neoplasm, but in a small proportion of biopsy material, these features may not be visible. In these cases, diagnosis may be difficult or even impossible.

An additional lesion that was considered in the differential diagnosis was a solitary fibrous tumor (SFT), which are rare spindle cell mesenchymal tumors [23]. The confirmation of this diagnosis is based on a positive result for CD34 and a negative result for S100 protein staining in immunohistochemistry [24]. 

Overall, the histopathological features and the immunoprofile were characteristic of a primary leiomyoma.

### 3.2. The Radiological Differential Diagnosis to Exclude Other Clival Diseases

Isolated clival lesions are rare pathologies that pose both diagnostic and therapeutic challenges, due to their clinical similarities. A diverse group of clival neoplasms can present with comparable clinical manifestations [25]. The symptoms include headaches and (less frequently) impaired vision or abducens (CN VI) nerve palsy [17]. The diagnosis of these pathologies should consider the clinical and radiological features of the tumors. The most common clival pathology is chordoma, which accounts for 40% of tumors within this area [25]. Differential diagnoses also include enchondromatous lesions, such as chondrosarcomas, fibrous dysplasia, myelomas, metastases and solitary fibrous tumors, then primary lymphomas and interosseous meningiomas, in order of their incidence in the clival area [26,27,28].

Chordomas are infrequent tumors that occur in the axial skeleton, which mainly affect the base of the skull and the sacrum. They usually appear in the third or fourth decade of life. The typical symptoms do not differ from those caused by other pathologies in this region. They probably originate from remnants of the fetal notochord, which are assumed to proliferate into chordomas [27]. Despite their rare incidence in the skeleton and as intracranial tumors, where they constitute 0.1–0.2% of lesions [25], they are the most common clival lesion [27], accounting for 23–40% of all lesions in this area [25]. Chordomas are characterized by slow growth, yet they have a high tendency toward local recurrence of up to 20% in the first year after diagnosis and local tissue invasion [29,30,31]. The prognostic imaging factors are not yet fully understood. 

On radiological images, chordomas are usually located centrally because they originate on the clivus. On MRI images, they show as homogeneous iso- or hyperintense lesions on T1-weighted imaging and heterogeneous hypo- or hyperintense lesions on T2-weighted imaging [8]. On non–contrast CT images, these lesions show as well-circumscribed, hypoattenuating, bone-destructive and heterogeneous. The most complicated differential diagnosis of clival chordomas is chondrosarcomas due to their similar radiological and histopathological features.

Although chordomas are rather rare among clival tumors (with an incidence rate of 0.08 per 100,000 [31]), they have been reported more frequently than primary leiomyomas [10]. 

Chondrosarcomas are locally aggressive tumors of mesenchymal origin and account for 6% of clival tumors. These lesions typically occur in the fourth decade of life. They are characterized by a very slow growth rate and therefore, can reach considerable sizes, while causing bone erosion and the displacement of the neural and vascular structures at the same time. Clival chondrosarcomas develop within synchondroses, especially within the petroclival synchondrosis, and involve the surrounding regions, such as the clivus, sphenoid sinuses and the middle and posterior cranial fossae, even extending to the upper cervical area. These tumors are characterized by low to intermediate signal intensity on T1-weighted imaging and high signal intensity in T2-weighted and FLAIR imaging [32,33]. 

Fibrous dysplasia (FD) is a benign, congenital skeletal disorder that is characterized by the replacement of normal bone with fibrous tissue. FD most often affects long bones [34]. Fibrous dysplasia is also characterized by a high frequency of activating mutations of the GNAS gene, which encodes the stimulatory G-protein αsubunit (Gsα) [35]. It has an incidence of 1 per 30,000 [36], but fibrous dysplasia in the clivus is a casuistic incident. There are three clinical patterns of these lesions. The vast majority of FD cases involve single bones, which are called monostotic cases. When several bones are involved, the term polyostotic is used. The third clinical pattern of FD is the craniofacial form. 

Fibrous dysplasia can occur at any age, but most cases are diagnosed around the age of 30 in both sexes equally [37]. FD is usually found accidentally due to the absence of clinical symptoms or the occurrence of nonspecific symptoms, such as headaches [38]. On T1-weighted imaging, FD is characterized by low to intermediate signal intensity and variable signal intensity on T2-weighted imaging [39]. The low T1-weighted signal intensity requires the differential diagnoses of chordomas, myelomas, chondrosarcomas, cholesteatomas, mucoceles and pituitary macroadenomas [40]. FD may also produce intermediate signal intensity on T1-weighted imaging and thus, can mimic soft tissue tumors. In addition, the strong contrast enhancement may resemble lymphomas, Langerhans cell histiocytosis or intraosseous hemangiomas [32]. 

The differential diagnosis of clival lesions also includes metastases on the occipital bone, which are very rare and are heterogenous in origin. Both can cause the initial symptoms of tumors or they may even develop after a long latency period of primary cancer. Clival metastases have been described in individual case reports. Metastases on the clivus have been reported from prostate cancer in most cases, while those from thyroid cancer and hepatocarcinoma have been reported less frequently. Metastases from Ewing’s sarcomas, melanomas and gastric and renal carcinomas have been described in single reports [25,27,32,40]. The probability of finding metastases on the clivus of patients with known primary malignancies is relatively high in contrast to those with unknown primary tumors. The presence of metastases seems less probable than finding a chordoma or chondrosarcoma [32]. The radiological features of metastases in the region of the clivus are nonspecific. In terms of differentiation between chordomas and chondrosarcomas, the most important aspect is hypointense signals on T2-weighted imaging [27]. 

We considered intracranial leiomyoma metastases separately, which are often referred to as benign metastasizing leiomyomas (BMLs). Metastases from leiomyomas to distant locations have usually been described solely in case reports or included in series with other tumors at the base of the skull [27]. Only a few cases of BMLs have been reported at the base of the skull [17,25]. BMLs are benign, slow-growing tumors, but they pose the potential risk of recurrence and malignancy [41]. The etiology of BMLs remains controversial [42]. They are usually associated with a previous history of uterine leiomyomas. BMLs in the brain are exceedingly rare, while the more common locations are the lungs, pelvis cavity and retroperitoneal space. Intracranial lesions have been described in single clinical images, including those in the region of the base of the skull [43] and the posterior parietal area. BMLs should be taken into consideration in the case of women of reproductive age who have a history of uterine leiomyomas or hysterectomy and are complaining of headaches or neurological deficits [41]. BMLs may even appear several years after surgery [43]. 

Our patient had no history of uterine leiomyomas, which was confirmed by ultrasonography, or hysterectomy, which allowed us to exclude the possibility of clival metastases from uterine fibroids.

Another uncommon occurrence in the area of the clivus is plasmacytomas, which are localized monoclonal neoplastic proliferations of plasma cells that occur in bone or soft tissue and account for less than 1% of head and neck cancers combined. Plasma cell tumors at the base of the cranium usually appear as a symptom of multiple myelomas [42], which are disseminated forms of plasma cell neoplasms. They can also present as localized forms, solitary plasmacytomas of bone (SPBs) or extramedullary plasmacytoma (EMPs) [44]. Only 20 cases of solitary intracranial plasmacytomas have been reported in the literature. Solitary intracranial plasmacytomas do not show any clinical symptoms for a long time. If symptoms appear, they mainly include headaches and symptoms resulting from the compression of nerve structures [45]. These lesions are isointense on T1-weighted imaging and moderately hyperintense in T2-weighted imaging. The multiple myeloma in the CT image has an invasive outline [32]. 

Hemangiopericytomas/solitary fibrous tumors (HPCs) were also taken into consideration. Previously, HPC/SFTs were classified separately as meningeal solitary fibrous tumors and hemangiopericytomas [46]. In the 2016 WHO Classification of Tumors of the Central Nervous System, HPCs were included in the same group as SFTs. HPC/SFTs are rare mesenchymal non-meningothelial tumors [47] that consist of Zimmermann pericytes and contractile mesenchymal perivascular cells [48]. Typically, SFTs are located in the visceral pleura, chest walls, sternum or abdominal cavity. SFTs are very rare within the CNS [20]. The intracranial locations for these tumors include the cerebellopontine angle area, supratentorial region, clivus, frontal lobes and brain ventricles [49]. HPCs have several clinical and imaging similarities to meningiomas [50]. Meningeal hemangiopericytomas account for fewer than 1% of brain tumors [38] and mainly affect adults between the fourth to the sixth decade of life [47]. Tumors with hemangiopericytoma phenotypes are characterized by a high recurrence rate of 90% over 12 years and extracranial metastases are observed in 93% of cases [50,51]. On MRI images, HPCs are isointense on T1-weighted imaging and hyperintense or mixed intensity on T2-weighted imaging. After contrast administration, the tumors have variable contrast enhancement. CT imaging shows solitary masses in the skull without calcification or hyperostosis [47].

Additionally, primary lymphomas at the base of the skull were included in the differential diagnosis. Primary lymphomas at the base of the skull are considered to be extremely rare tumors and are infrequently encountered in clinical practice [52,53,54]. According to the WHO Classification of Tumors, primary CNS lymphomas (PCNSLs) account for 2.4–3% of brain tumors and about 4–6% of extranodal lymphomas [52]. These tumors most commonly occur in the paratentorial area. Lymphomas at the base of the skull have usually been described in single case reports in the literature [53]. These lymphomas mainly affect patients between 50 and 70 years old [37,42] and typically occur in patients who already have lymphomas in the lymph nodes [53]. Differentiating between lymphomas from other lesions at the base of the skull (including leiomyomas) is challenging due to the nonspecific imaging. They are often solitary tumors that infiltrate bones. If the base of the cranium is affected, the symptoms of the lymphomas are not pathognomonic [54]. On pre-contrast CT imaging, lymphomas at the base of the skull present as iso- or hyperdense lesions. Their signal intensity on MRI images is nonspecific. On T1-weighted imaging, lymphomas at the base of the skull are iso- or hypointense and on T2-weighted imaging, they are iso- or hyperintense [53]. The tumors are enhanced after contrast administration. 

The area of the clivus can also be affected by interosseous meningiomas (IOMs). IOMs are extradural meningiomas that originate from bone [47] and account for 0.6–1.2% of all meningiomas. Extradural meningioma is the term used for lesions in other locations, such as the skin or neck. IOMs develop from convexity meningiomas or tumors at the skull base. They are slow-growing tumors and can affect neural structures [55]. Clival IOMs are extremely rare lesions, while more common locations include spheno-orbital, petrous and even anterior skull regions. A characteristic feature of IOMs is their slow and asymptomatic growth. Cranial nerve involvement is also possible. Osteoblastic, osteolytic and mixed types of IOMs can be distinguished radiologically. On T1-weighted MRI imaging, these tumors are isointense and are enhanced after contrast administration. In contrast, the lesions are hyperintense on T2-weighted imaging [56]. 

## 4. Conclusions

According to our report, this case of a primary leiomyoma on the clivus of an immunocompetent patient is the first to be reported in the literature. To date, leiomyomas have only been seen in this location in a patient with HIV [15] and two other patients with unknown immune statuses [8,12]. Single clival lesions are extremely uncommon and have heterogeneous origins. Based on epidemiological, clinical, radiological and histopathological data, appropriate clinical interventions should be performed [57]. Primary interosseous leiomyomas on the clivus should also be suggested in differential diagnoses after more common lesions for this area (such as chordoma, chondrosarcoma and metastases) have been excluded [22]. The differentiation between primary leiomyomas and benign metastatic leiomyomas (BMLs) should also be considered [28,41,42], especially when there are risk factors for fibroids. This review could provide new information on the pathogenesis of fibroids and their ability to spread to distant organs; however, more case reports of this phenomenon need to be collected.

## Figures and Tables

**Figure 1 diagnostics-13-00009-f001:**
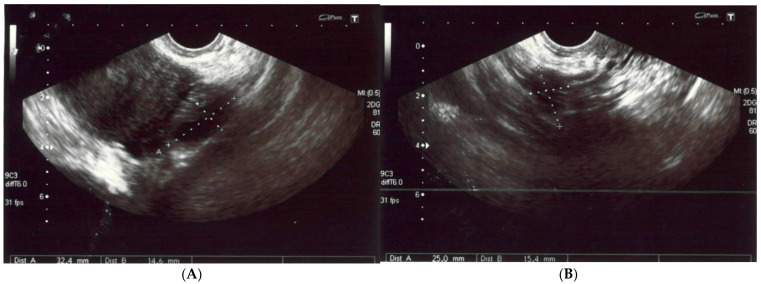
(**A**,**B**) The left and right ovaries, with their diameters. The right ovary measured 25 × 15 mm and the left ovary 24 × 14 mm. (**C**) The transverse plan of the uterus at its widest dimensions. In this plane, the uterine width (Ut-W) was measured. The uterus was of normal size in anterior flexion. (**D**) The midsagittal plane of the uterus showing the uterine fundus, myometrium, endometrium (6 mm thick), isthmus, cervix and cul-de-sac.

**Figure 2 diagnostics-13-00009-f002:**
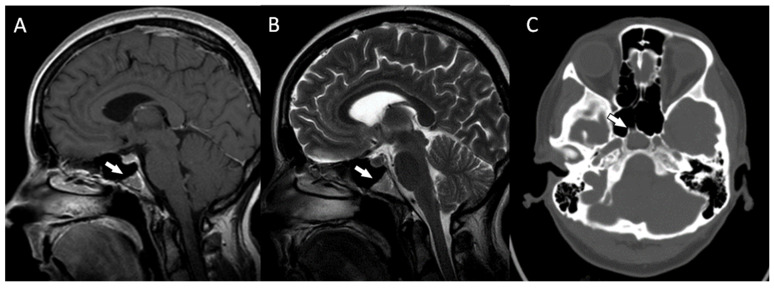
(**A**) The magnetic resonance postcontrast T1-weighted sagittal image, demonstrating the isointense lesion with hyperintense borders (arrow). The image suggested chordoma or fibrous dysplasia (**B**). The magnetic resonance T2-weighted dark fluid sagittal image, showing the isointense clival lesion (arrow) (**C**). The computed tomography image, showing the centrally located lesion with a hypersclerotic rim (arrow). The tumor did not infiltrate the sphenoid sinuses.

**Figure 3 diagnostics-13-00009-f003:**
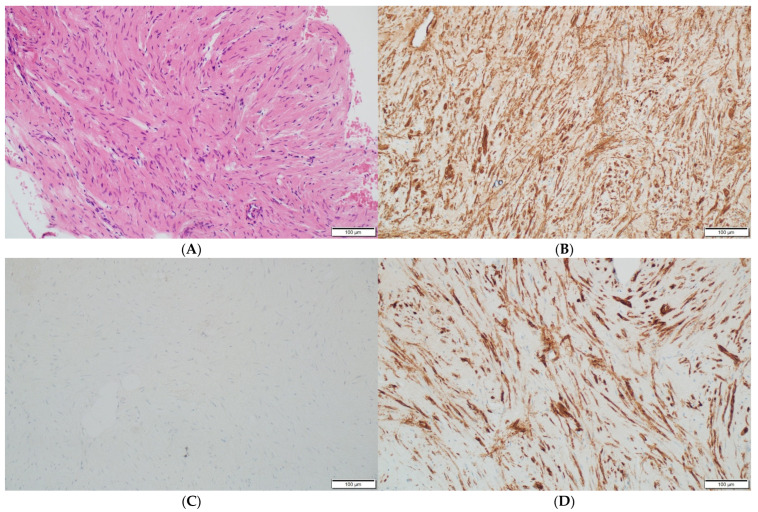
(**A**) The leiomyoma is composed of cells with elongated nuclei without cellular atypia (hematoxylin and eosin staining at a magnification of ×40 and ×200). (**B**) The immunohistochemistry for h-Caldesmon, which is positive in the leiomyoma cells (at a magnification of ×200). (**C**) The immunohistochemical staining for nuclear marker beta-catenin, which is negative in the leiomyoma cells (at a magnification of ×200). (**D**) The immunohistochemistry for smooth muscle antigen (SMA), which shows reactivity (at a magnification of ×200).

**Figure 4 diagnostics-13-00009-f004:**
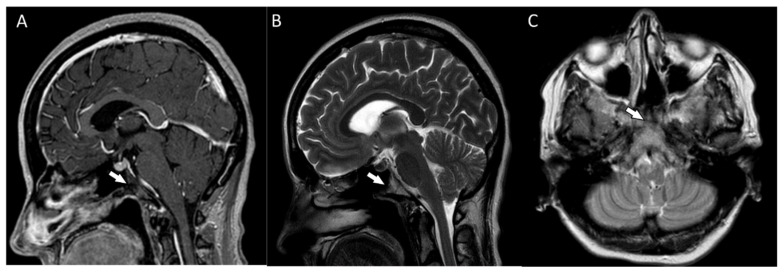
(**A**,**B**) The postoperative magnetic resonance T1-weighted and T2-weighted sagittal images and the postoperative magnetic resonance T2-weighted coronal image (**C**) showing some minimal residual lesions within the clival region (arrows).

## Data Availability

Not applicable.

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
