# Peer review of "A Rare Case of a Primary Leiomyoma of the Clivus in an Immunocompetent Patient and a Review of the Literature Regarding Clival Lesions"

_diagnostics, 2022, doi:10.3390/diagnostics13010009_

Round 1

Reviewer 1 Report (Previous Reviewer 3)

The authors resolved my concerns.

Author Response

Natalia Rzewuska

Department of Gynecological Endocrinology,

Medical University of Warsaw, Poland

                                                                                              December 12th, 2022

Subject: Revision of the manuscript ID diagnostics-1852535.

Title: A rare case of a primary leiomyoma of the clivus in an immunocompetent patient and a review of the literature regarding clival lesions

Diagnostics

Dear Reviewer,

Thank you for all your comments that have improved our manuscript. The work was changed thanks to your suggestions and has undergone English language editing by MDPI.

We hope that the current format of the work can be published.

We are grateful for Your time.

Yours sincerely,

Natalia Rzewuska

the corresponding author

Reviewer 2 Report (Previous Reviewer 1)

All of my comments were addressed. 

Author Response

Natalia Rzewuska

Department of Gynecological Endocrinology,

Medical University of Warsaw, Poland

                                                                                              December 12th, 2022

Subject: Revision of the manuscript ID diagnostics-1852535.

Title: A rare case of a primary leiomyoma of the clivus in an immunocompetent patient and a review of the literature regarding clival lesions

Diagnostics

Dear Reviewer,

Thank you for all your comments that have improved our manuscript. The work was changed thanks to your suggestions and has undergone English language editing by MDPI.

We hope that the current format of the work can be published.

We are grateful for Your time.

Yours sincerely,

Natalia Rzewuska

the corresponding author

Reviewer 3 Report (New Reviewer)

In this manuscript, the authors present a case report describing a middle-aged (56-year-old) apparently HIV-free patient who developed a primary leiomyoma localized to the clivus. Overall, the paper is well written and worthy of publication, but I ask the authors to provide more details that can clearly verify that the patient did not suffer from any immunodeficiency. Additionally, I ask the authors to clearly explain whether immunohistochemistry or other techniques were performed for EBV in this patient.

Review of the manuscript titled “A rare case of a primary leiomyoma of the clivus in an immunocompetent patient and review of the literature regarding clival lesions”.

Abstract: Line 25, please correct: Desmin, not demin!

Introduction: this section is complete and clear. Because the authors mentioned the involvement of gastrointestinal system, please add this paper about a rare localization of a leiomyoma in the liver.

Colagrande, A.; Cazzato, G.; Fedele, S.; Andriola, V.; Ingravallo, G.; Resta, L.; Vincenti, L. A Unique Case of the Transformation of a Hepatic Leiomyoma into Leiomyosarcoma with Pancreatic Metastases: Review of the Literature with Case Presentation. Reports 2022, 5, 2. https://doi.org/10.3390/reports5010002

Case Report: the section is well written. Please, I have few suggestions to improve the quality of this manuscript:

1.      Add some arrows to the radiological picture showing the mentioned lesions.

2.      Use always a “time”: in the captions of the figure, use the past simple and not the present simple.

Discussion: Line 136: the authors write :” Single cases concern basal ganglia, and occipital lobe [8,11,12].”. Why is there a comma? It’s not necessary! Please, correct.

My recommendation is: minor revisions.

Review of the manuscript titled “A rare case of a primary leiomyoma of the clivus in an immunocompetent patient and review of the literature regarding clival lesions”.

Abstract: Line 25, please correct: Desmin, not demin!

Introduction: this section is complete and clear. Because the authors mentioned the involvement of gastrointestinal system, please add this paper about a rare localization of a leiomyoma in the liver.

Colagrande, A.; Cazzato, G.; Fedele, S.; Andriola, V.; Ingravallo, G.; Resta, L.; Vincenti, L. A Unique Case of the Transformation of a Hepatic Leiomyoma into Leiomyosarcoma with Pancreatic Metastases: Review of the Literature with Case Presentation. Reports 2022, 5, 2. https://doi.org/10.3390/reports5010002

Case Report: the section is well written. Please, I have few suggestions to improve the quality of this manuscript:

1.      Add some arrows to the radiological picture showing the mentioned lesions.

2.      Use always a “time”: in the captions of the figure, use the past simple and not the present simple.

Discussion: Line 136: the authors write :” Single cases concern basal ganglia, and occipital lobe [8,11,12].”. Why is there a comma? It’s not necessary! Please, correct.

My recommendation is: minor revisions.

Author Response

Natalia Rzewuska

Department of Gynecological Endocrinology,

Medical University of Warsaw, Poland

                                                                                                              December 12th, 2022

Subject: Revision of the manuscript ID diagnostics-1852535.

Title: A rare case of a primary leiomyoma of the clivus in an immunocompetent patient and a review of the literature regarding clival lesions

Diagnostics

Dear Reviewer,

Thank you for the opportunity to correct our paper. We value your contribution to the review of our article. We introduced the appropriate changes and followed your suggestions.

Comment 1)  “Overall, the paper is well written and worthy of publication, but I ask the authors to provide more details that can clearly verify that the patient did not suffer from any immunodeficiency.”

Answer 1) Thank you very much for your comment. As suggested, we added some more information about the patients. We included it in lines 76 – 78.

Comment 2) “Additionally, I ask the authors to clearly explain whether immunohistochemistry or other techniques were performed for EBV in this patient.”

Answer 2) In order to diagnose EBV, immunohistochemistry was performed. A negative result was obtained. We have made the mentioned change in the manuscript.

Comment 3) “Line 25, please correct: Desmin, not demin!”

Answer 3) We provide the needed change in line 25.

Comment 4) “Introduction: this section is complete and clear. Because the authors mentioned the involvement of the gastrointestinal system, please add this paper about a rare localization of a leiomyoma in the liver.

Colagrande, A.; Cazzato, G.; Fedele, S.; Andriola, V.; Ingravallo, G.; Resta, L.; Vincenti, L. A Unique Case of the Transformation of a Hepatic Leiomyoma into Leiomyosarcoma with Pancreatic Metastases: Review of the Literature with Case Presentation. Reports 2022, 5, 2. https://doi.org/10.3390/reports5010002”

Answer 4) We appreciate the very insightful comment, for this reason, we have included the information about hepatic leiomyoma in line 35.

Comment 5)

“1.      Add some arrows to the radiological picture showing the mentioned lesions.

  1. Use always a “time”: in the captions of the figure, use the past simple and not the present simple.”

Answer 5) We inserted the arrows both in the radiological image before surgery and after surgery. Thank you again for the important remark. We used the proper time.

Comment 6) “Discussion: Line 136: the authors write:” Single cases concern basal ganglia, and occipital lobe [8,11,12].”. Why is there a comma? It’s not necessary! Please, correct.”

Answer 6) We have corrected the sentence as suggested.

We have made corrections to the main text and the changes are marked in red.

In addition, our manuscript has undergone English language editing by MDPI.

We are grateful to the Reviewer for Your time and constructive comments on our manuscript, and we hope the revised version is ready for publication now and look forward to hearing from you at the right time.

Yours sincerely,

Natalia Rzewuska

the corresponding author

This manuscript is a resubmission of an earlier submission. The following is a list of the peer review reports and author responses from that submission.

Round 1

Reviewer 1 Report

In this case report & literature review study, the authors reported a case of leiomyoma in the area of clivus, in which the tumor tissue manifested as  typical smooth muscle cells positively stained with SMA, while negatively for S100, beta-catenin, and PGR. It is an interesting case report with systemic review, although the intracranial leiomyoma is rare, several case reports and literature reviews had been published, and the causal relationship between immune-incompetence (such as HIV or EBV infection) and intracranial leiomyoma is not confirmed. Based on the published literatures, the incidence of intracranial leiomyoma between immune-incompetent and immune-competent patients is not differed. The location of leiomyoma in the area of clivus did not show the particularity compared to other types of intracranial leiomyoma. In addition, some major concerns can not be ignored regarding to the diagnosis, differential diagnosis, and clinical recommendation.   

Line 231, the authors stated that that uterine leiomyoma was excluded by ultrasonography or hysterectomy, it is better to provide the ultrasonography results to testify their diagnosis. In addition, the hysterectomy history was not mentioned in the case illustration,  the authors need to confirm if hysterectomy was performed or not, and the pathological inspection results should be presented if the patient had hysterectomy.  

Although uterine leiomyoma is the most common type of leiomyoma, and it can be diagnosed by ultrasonography,  leiomyoma can also be found in gastrointestinal tracts, skin, and other tissues. In this case, the authors only stated that the uterine leiomyoma history was excluded by ultrasonography, the potential risk of leiomyoma diagnosis in other organs was not excluded, based on the case illustration. Therefore, the diagnosis of primary leiomyoma in the clivus is not accurate.

The first-ever reported case of primary leiomyoma of the clivus in an immunocompetent patient is the highlight of this study. In fact, clivus leiomyoma had been reported in patients with HIV infection or with unknown/immunocompromised status. Again, there is no clear evidence show that intracranial leiomyoma is related to immune-incompetence, the intracranial leiomyoma had been reported in young and immunocompetent patients (aged 17 to 50). In addition, the patient in this study was 56 years old, the health status evaluation was not provided in the case illustration. Taken together, the highlights of this study are not matched with the contents.

Reviewer 2 Report

there are no specific comments

Reviewer 3 Report

The authors report a rare case of intracranial leiomyoma in the clivus region in an immunocompetent patient. As the authors stated, 30 reports were reported to time in various regions of the cranial vault.

The report is written a little unusual, and needs some revisions before achieving the acceptable form, while the case itself is well noted.

The introduction should be reorganised into paragraphs on leiomyoma in general, intracranial presentation and etiology in intracranial cases, followed by the introductory statement to your case report, as a final paragraph of the Introduction.

A report section is well written, however it needs some reorganisation. Please merge figures 1, 2 and 3 to 1 A.B.C. Also do the same with 5, 6, 7 and 8. 5 may be excluded, as it brings no novel details other than the Fig. 5.

There should be contrast enhanced MRI images from both preoperative and postoperative state. These should be also comparable, presented with the same or near same slices in axial, sagittal and eventually coronal planes.

The discussion is poor, as it only discusses differential diagnosis, which should only be the part of the discussion, therefore needs the text to be shortened (the HP might even be presented as a Table). First paragraph should summarise your case with its unique features. which should be discussed afterwards, and compared to the similar cases (intracranial leiomyoma) previously reported.